# A Combinatorial Perspective on Transfer Learning

**Jianan Wang**    **Eren Sezener**    **David Budden**    **Marcus Hutter**    **Joel Veness**

DeepMind

`aixi@google.com`

## Abstract

Human intelligence is characterized not only by the capacity to learn complex skills, but the ability to rapidly adapt and acquire new skills within an ever-changing environment. In this work we study how the learning of modular solutions can allow for effective generalization to both unseen and potentially differently distributed data. Our main postulate is that the combination of task segmentation, modular learning and memory-based ensembling can give rise to generalization on an exponentially growing number of unseen tasks. We provide a concrete instantiation of this idea using a combination of: (1) the Forget-Me-Not Process, for task segmentation and memory based ensembling; and (2) Gated Linear Networks, which in contrast to contemporary deep learning techniques use a modular and local learning mechanism. We demonstrate that this system exhibits a number of desirable continual learning properties: robustness to catastrophic forgetting, no negative transfer and increasing levels of positive transfer as more tasks are seen. We show competitive performance against both offline and online methods on standard continual learning benchmarks.

## 1 Introduction

Humans learn new tasks from a single temporal stream (online learning) by efficiently transferring experience of previously encountered tasks (continual learning). Contemporary machine learning algorithms struggle in both of these settings, and few attempts have been made to solve challenges at their intersection. Despite obvious computational inefficiencies, the dominant machine learning paradigm involves i.i.d. sampling of data at massive scale to reduce gradient variance and stabilize training via back-propagation. In the case of continual learning, the batch i.i.d. paradigm is often further extended to sample from a memory of experiences from all previous tasks. This is a popular method of overcoming "catastrophic forgetting" [CG88, MC89, Rob95], whereby a neural network trained on a new target task rapidly loses in its ability to solve previous source tasks.

Instead of considering "online" and "continual" learning as inconvenient constraints to avoid, in this paper we describe a framework that leverages them as desirable properties to enable effective, data-efficient transfer of previously acquired skills. Core to this framework is the ability to ensemble task-specific neural networks at the level of individual nodes. This leads naturally to a desirable property we call *combinatorial transfer*, where a network of $m$ nodes trained on $h$ tasks can generalize to $h^m$ "pseudo-tasks". Although the distribution of tasks and pseudo-tasks may differ, we show that this method works very well in practice across a range of online continual learning benchmarks.

The ability to meaningfully ensemble individual neurons is not a property of contemporary deep neural networks, owing to a lack of explicit modularity in the distributed and tightly coupled feature representations learnt via backpropagation [BDR$^+$19, CFB$^+$19, PKRCS17]. To concretely instantiate our learning framework we instead borrow two complementary algorithms from recent literature: the Gated Linear Network (GLN) and the Forget-Me-Not Process (FMN). We demonstrate that properties of both models are complementary and give rise to a system suitable for online continual learning.

## 2 Background

*Transfer learning* can be defined as the improvement of learning a new (target) task by the transfer of knowledge obtained from learning about one or more related (source) tasks [TS09]. We focus on a generalization of transfer learning which has been studied under the names *continual learning* [PKP$^+$19] or lifelong-learning [CL18]. Rather than considering just a single source and target task, continual learning considers a sequence of tasks drawn from a potentially non-stationary distribution. The aim is to learn future tasks while both (1) not forgetting how to solve past tasks, and ideally (2) leveraging similarities in previous tasks to improve learning efficiency. Despite the similarity with online learning, this sequence of tasks is rarely exposed as a temporal stream of data in the literal sense. We refer to the intersection of these challenges as *online continual learning*.

A long-standing challenge for neural networks and continual learning is that the performance of a neural network on a previously encountered source task typically rapidly degrades when it is trained on a new target task. This phenomenon is widely known as *catastrophic forgetting* [CG88, MC89, Rob95], but we will sometimes refer to it as *negative backward transfer* to distinguish it from *positive forward transfer* (source task training improving target task performance) and *positive backward transfer* (target task training improving source task performance).

Previous approaches for overcoming catastrophic forgetting typically fall into two categories. The first category involves replaying source task data during target task training [Rob95, RKSL17, LPR17, CRE$^+$19, ABT$^+$19]. The second category involves maintaining task-specific network weights that can be leveraged in various ways [DJV$^+$13, GDDM14], typically involving regularization of the training loss [KPR$^+$17, ZPG17, SLC$^+$18]. These methods share some combination of the following limitations: (1) requiring knowledge or accurate estimation of task boundaries; (2) additional compute and memory overhead for storing task-specific experience and/or weights; and (3) reliance on batch i.i.d. training that leads to unsuitability for online training. A desirable algorithm for "human-like" continual learning is one that addresses these limitations while simultaneously encouraging positive forward and backward transfer.

## 3 Combinatorial Transfer Learning

We start with a short thought experiment. Consider two neural networks trained on two different tasks, as shown in Figure 1. If the networks are modular in the sense that we could *locally* (per node/neuron) ensemble the two networks, one could reasonably expect a global ensemble to work beyond the range of tasks it was trained on by leveraging its previously learnt building blocks across the two tasks. Now consider a network with $m$ nodes, independently trained on $h$ tasks, resulting in $h$ sets of parameters. If we locally ensemble these $h$ networks, such that for each position in the ensemble network we can pick a node *of same position* from any of the networks, we would have $h^m$ possible instantiations. We refer to the task solved by each instantiation as a *pseudo-task* and claim that such a system exhibits combinatorial transfer.

There are two issues with this thought experiment. First, contemporary neural networks do not exhibit the modularity necessary for local ensembling. Many recent studies aim at identifying reusable modular mechanisms to support causal and transfer learning [BDR$^+$19, CFB$^+$19, PKRCS17], but this is made fundamentally difficult by the nature of the distributed and tightly coupled representations learnt via backpropagation. Second, it is not obvious whether the distribution of pseudo-tasks (ensembles of partial solutions to source tasks) meaningfully captures the complexities of the unseen target tasks in a useful fashion. To address these issues, we seek a solution that satisfies the following desiderata: (1) associated to every node is a learning algorithm with its own local loss function; and (2) the nodes coordinate, in the sense that they work together to achieve a common goal. In Section 4 we propose an algorithm that fulfills these requirements. In Section 5 we evaluate its performance on a number of diagnostic and benchmark tasks, demonstrating that the combinatorial transfer property is present and leads to improved continual learning performance compared with previous state-of-the-art online and offline methods.

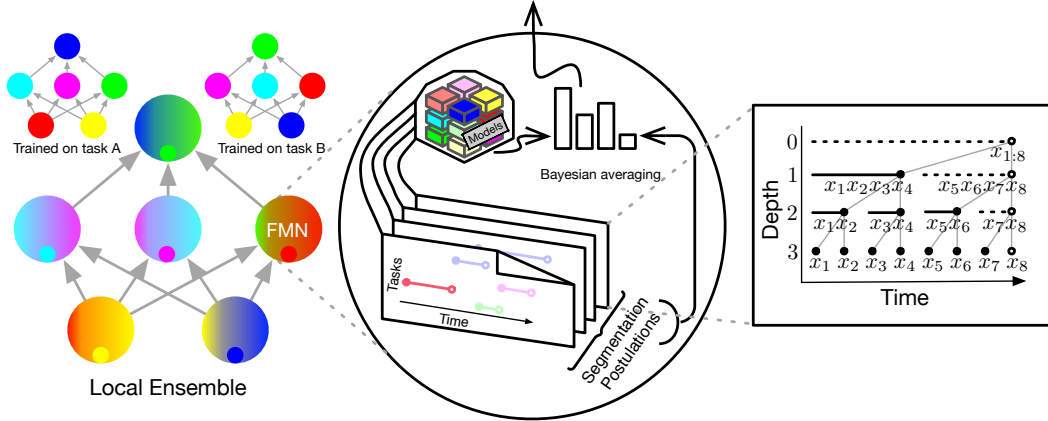

Figure 1: Neural Combinatorial Transfer Learning (NCTL). (**Left**) Consider two 6-node networks trained separately on tasks A and B. These networks can be locally ensembled (keeping node positions fixed) to form $2^6$ new networks. One particular instantiation is shown with small nodes placed within each (large) node of the local ensemble. Rather than ensembling by selecting models/nodes, we perform Bayesian Model Averaging, which is indicated using the colour gradients. (**Center**) Each NCTL node uses the FMN Process as its ensembling technique. Best solutions identified by the GGM base model are stored in a model pool and combined via Bayesian Model Averaging. Further (hierachical) Bayesian Model Averaging is performed over postulationed task segmentations, each of which is a temporal partition of the input stream into task segments. (**Right**) A candidate task segmentation can be any pruning of the tree structure shown, with the segmentation described by the set of leaves.

# 4 Algorithm

In this Section we describe the Neural Combinatorial Transfer Learning (NCTL) algorithm, a concrete instantiation (see Figure 1) of the above ideas using (1) Gated Geometric Mixer (GGM) as the node-level modular learning algorithm; and (2) Forget-Me-Not Process (FMN) for automatic task segmentation and local ensembling of the learnt GGM solutions.

**Gated Geometric Mixer.**    Geometric Mixing is a well studied ensemble technique for combining probabilistic forecasts [Mat12, Mat13]. Given $\boldsymbol{p} := (p_1, \ldots, p_K)$ input probabilities predicting the occurrence of a single binary event, geometric mixing predicts probability $p' := \sigma(\boldsymbol{w}^\top \cdot \boldsymbol{\sigma}^{-1}(\boldsymbol{p}))$, where $\sigma(x) := 1/(1 + \mathrm{e}^{-x})$ denotes the sigmoid function, $\sigma^{-1}$ defines the logit function, and $\boldsymbol{w} \in \mathbb{R}^K$ is the weight vector which controls the relative importance of the input forecasts.

A Gated Geometric Mixer (GGM) [VLB$^+$19] is the combination of a gating procedure and geometric mixing. Gating has the intuitive meaning of mapping particular input examples to a particular choice of weight vector for use with geometric mixing. The key change compared with normal geometric mixing is that now our neuron will also take in an additional type of input, *side information $z \in \mathcal{Z}$*, which will be used by the contextual gating procedure to determine an active subset of the neuron weights to use for a given example. Associated with each GGM is a context function $c : \mathcal{Z} \to \mathcal{C}$, where $\mathcal{C} = \{1, \ldots, C\}$ for some $C \in \mathbb{N}$ is the *context space*. Each GGM is parameterized by a matrix $\mathbf{W} = [\mathbf{w}_1^\top, ..., \mathbf{w}_k^\top]^\top$ with row vector $\mathbf{w}_i \in \mathbb{R}^K$. The context function $c$ is responsible for mapping a given piece of side information $z \in \mathcal{Z}$ to a particular row $\mathbf{w}_{c(z)}$ of $\mathbf{W}$, which we then use with standard geometric mixing. Formally, $\mathrm{GGM}_{c,\mathbf{W}}(\boldsymbol{p}, z) := \sigma(\mathbf{w}_{c(z)} \cdot \boldsymbol{\sigma}^{-1}(\boldsymbol{p}))$. One can efficiently adapt the weights $\mathbf{W}$ online using Online Gradient Descent [Zin03] by exploiting convexity under the logarithmic loss as described in [VLB$^+$19].

**Forget-Me-Not Process.**    Generalizing stationary algorithms to non-stationary environments is a key challenge in continual learning. In this work we choose to apply the Forget-Me-Not (FMN) process [MVK$^+$16] as our choice of node-level ensembling technique. The FMN process is a probabilistic meta-algorithm tailored towards non-i.i.d., piecewise stationary, repeating sources. This

meta-algorithm takes as input a single base measure $\rho$ on target strings and extends the Partition Tree Weighting algorithm [VWBG13] to incorporate a memory of up to $k$ previous model states in a data structure known as a model pool. It efficiently applies Bayesian model averaging over a set of postulated segmentations of time (task boundaries) and a growing set $\mathcal{M}$ of stored base model states $\rho(\cdot|s)$ for some subsequence $s$ of $x_{1:n}$, while providing a mechanism to either learn a new local solution or adapt/recall previous learnt solutions. Here our base measure $\rho$ will be (implicitly) defined from a sequential application of GGM, and formally defined later.

The FMN algorithm is derived and described in [MVK$^+$16]. It computes the probability $p' = \text{FMN}_d(x_{1:n}) \in [0; 1]$ of a string of binary targets $x_{1:n} \in \{0, 1\}^n$ of length $n$, e.g. $x_t$ could be a binary class label (of feature $z_t$ introduced later). For this, it hierarchically Bayes-mixes an exponentially large self-generated class of models from a base measure $\rho$ in $O(kn \log n)$ time and $O(k \log n)$ space, roughly as follows: For $n = 2^d$ and for $j = 0, .., d$, it breaks up string $x_{1:n}$ into $2^j$ strings, each of length $2^{d-j}$, which conceptually can be thought of in terms of a complete binary tree of depth $d$ (see Figure 1, right). For each substring $x_{a:b}$, associated to each node of the tree will be a probability $\xi(x_{a:b})$ obtained from a Bayesian mixture of all models in the model pool $\mathcal{M}_a$ at time $a$. Taking any (variable depth) subtree (which induces a particular segmentation of time), concatenating the strings at its leaves gives back $x_{1:n}$, therefore the product of their associated mixture probabilities gives a probability for $x_{1:n}$. Doing and averaging this (see [MVK$^+$16, Eq.7]) for all possible $O(2^n)$ subtrees, which can be done incrementally in time $O(k \log n)$ per string element, gives $\text{FMN}_d(x_{1:n}|\rho)$.

The models in the model pool are generated from an arbitrary adaptive base measure $\rho$ by conditioning it on past substrings $x_{a:b}$. For example, $\rho$ could be a Beta-Bernoulli model whose weights are updated using Bayesian inference, or something more sophisticated such as a GGM. At time $t$, $\mathcal{M}_t$ contains at most $k$ "versions" of $\rho$, with $\mathcal{M}_1 := \{\rho\}$. For $t = 2, ..., n$, the path leading to the $t$th leaf of the tree in Figure 1 (right) is traversed; whenever a string $x_{a:b}$ with $b = t$ is encountered, then the model $\rho^* \in \mathcal{M}_a$ assigning the highest probability to the node's string $x_{a:b}$ is either added to the model pool, i.e. $\mathcal{M}_{t+1} = \mathcal{M}_t \cup \{\rho^*(\cdot|x_{a:b})\}$, or ignored based on a Bayesian hypothesis test criterion given in [MVK$^+$16].

**Neural Combinatorial Transfer Learning (NCTL).**   We now instantiate the Combinatorial Transfer Learning framework. This will involve defining a feed-forward network structure exactly the same as for a GLN, but replacing the basic notion of neuron, a GGM, with the more intricate FMN process applied to a GGM. Informally, the output of GGM is first piped through FMN before being used in the NCTL neuron, so that each node is now well-suited to non-stationary continual learning. We use random half-space gating [VLB$^+$17, SHB$^+$20] to generate the gates for each neuron.

A GLN is a network of GGMs with two types of input to each node: the first is side information $z \in \mathcal{Z}$, which can be thought of as the input features in a standard supervised learning setup; the second is the input $\boldsymbol{p} \in [0; 1]^K$ to the node, which are the predictions output by the previous layer, or in the case of layer 0, $\boldsymbol{p}_0 := \boldsymbol{f}(z)$ for some function $\boldsymbol{f}$ of the side information $z$. Upon receiving an input $\boldsymbol{p}$ and side information $z$, each node attempts to directly predict the target probability $P[x_t = 1|z_t]$. Formally, neuron $j$ in layer $i$ outputs $q_{ij}^t := \text{GGM}_{c_{ij}(\cdot), \mathbf{W}_{ij}}(\boldsymbol{p}_{i-1}^t, z_t)$. A GLN would now feed $\boldsymbol{p}_i^t := \boldsymbol{q}_i^t := (q_{i1}^t, ..., q_{iK_i}^t)$, where $K_i$ is the number of neurons in layer $i$, as input to the next layer.

NCTL works in the same way but instantiates $\boldsymbol{p}_i^t$ differently. Formally, $q_{ij}$ determines base measures $\rho_{ij}$ on the target string $x_{1:n}$ by $\rho_{ij}(x_t = 1|x_{<t}) := q_{ij}^t$, where all dependencies (on $z, c, \mathbf{W}$ and previous layers) have been suppressed, and $\rho_{ij}(x_{1:n}) := \prod_{t=1}^n \rho_{ij}(x_t|x_{<t})$. Now $p_{ij}$ is determined by $p_{ij}^t := \text{FMN}_d(x_t = 1|x_{<t}; \rho_{ij})$, where $\text{FMN}_d(x_t|x_{<t}; \rho) := \text{FMN}_d(x_{1:t}|\rho) / \text{FMN}_d(x_{<t}|\rho)$ is the probability for $x_t$ predicted by FMN based on $\rho$. Finally, $\hat{P}[x_t = 1|z_t] = p_{L1}^t$ is the output of the top layer $L$ neuron, which is taken as the output of NCTL.

**Computational Complexity**   Assuming a fixed network structure and a constant-sized memory pool of $k$ items, the time and space complexity of NCTL to process $n$ symbols is $O(nk \log n)$ and $O(k \log n)$ respectively. Furthermore, the per sample running time is bounded by $O(k \log n)$.

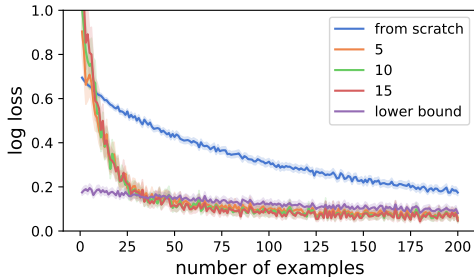
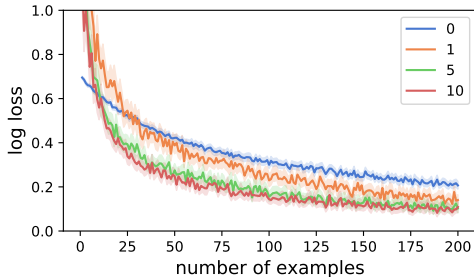

Figure 2: Demonstration of positive backward transfer with NCTL, averaged across 500 task sequences with unknown boundaries and identities. A model trained on increasing numbers of distractor tasks is rapidly ($\approx 30$ steps) able to match lower bound performance, and tends to *improve* on it, implying positive backward transfer.

Figure 3: Demonstration of positive forward transfer with NCTL, averaged across 500 task sequences with unknown boundaries and identities. The efficiency with which NCTL learns a randomly sampled held-out target task improves monotonically in the number of source tasks, implying positive forward transfer.

# 5 Experimental Results

We now explore the properties of the NCTL algorithm empirically. We present our analysis in three parts: in Section 5.1, we demonstrate that NCTL exhibits combinatorial transfer using a more challenging variant of the standard Split MNIST protocol; in Section 5.2, we compare the performance of NCTL to many previous continual learning algorithms across standard Permuted and Split MNIST variants, using the same test and train splits as previously published; in Section 5.3, we further evaluate NCTL on a widely used real-world dataset *Electricity* (Elec2-3) which exhibits temporal dependencies and distribution drift. We use feature vector of $K_0 = 784$ dimensions for MNIST (flattened) and $K_0 = 5$ dimensions for *Electricity*. Each feature vector is standardized component-wise to have zero mean and unit variance. This normalized feature vector is broadcast to every neuron as side information $z$. The inputs to each neuron are the predicted probabilities output by each neuron in the preceding layer, with the exception of layer 0, which takes $\boldsymbol{p}_0 = \sigma(z)$ as base predictions.

## 5.1 Evaluating Online Combinatorial Transfer

In Section 3 we proposed a framework by which a network with $m$ nodes trained on $h$ tasks will generalize to $h^m$ pseudo-tasks via local ensembling. What remains to be demonstrated is whether these pseudo-tasks are meaningful, i.e. whether combinatorial transfer is empirically beneficial for continual learning. We do this by measuring both *forward transfer* and *backward transfer* in an online regime, from a single stream of experience where both task boundaries and identities are unknown to the learning algorithm. We report performance in terms of instantaneous log-loss.

Most recent studies on continual learning define a protocol that makes use of an underlying MNIST (or similar) classification dataset. The popular (Disjoint) Split MNIST [ZPG17] involves separating the 10-class classification problem with 5 binary classification tasks, digits 0-vs-1, 2-vs-3 and so on. To assess combinatorial transfer, we propose a more challenging variant of the Disjoint Split MNIST protocol that includes two additional modifications. First, data is presented online (i.e. a single temporal stream) with random task durations (number of examples) equal to $100 + X$, where $X \sim \text{Geometric}(0.01)$, and no explicit signalling of task boundaries. Second, we generate tasks representing all unique pairs of different MNIST digits following the convention that the smaller digit is assigned to label 0. This protocol yields 45 different tasks. Some of these tasks will naturally conflict, for example in the case of 1-vs-3 versus 3-vs-5, notice that digit 3 has changed class label between tasks. From here onwards we refer to this protocol as *Free Split MNIST*.

**Positive Backward Transfer.** Catastrophic forgetting (negative backward transfer) is one of the most widely studied topics in continual learning. Here we measure backward transfer by first training NCTL on a randomly sampled task, followed by a varying number of *distractor tasks*, before finally evaluating on the initial task. Figure 2 shows the results. It is important to note that NCTL is not provided with any knowledge of task information (including task boundaries and task identities). This

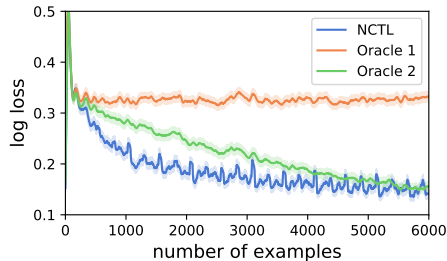

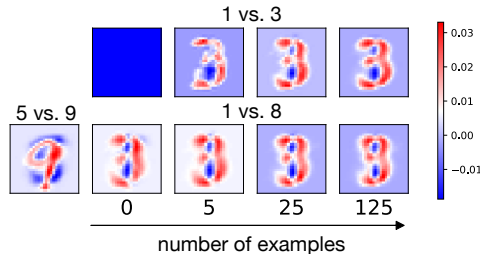

Figure 4: Average NCTL performance (1000 seeds) on Free Split MNIST compared to two GLN-based Oracles, with knowledge of task boundaries (1) and task identities (2). The ability of NCTL to outperform Oracle 2 is direct evidence for positive forward transfer and its capability to rapidly identify good solutions online.

Figure 5: Saliency maps for sequentially training NCTL on task 1-vs-3, 5-vs-9 and 1-vs-8. (**Top**) shows rapid learning of task 1-vs-3 as the characteristic shape of digits emerges. (**Bottom**) shows that when task transitions from 5-vs-9 to 1-vs-8 NCTL rapidly recovers learnt solution for qualitatively useful task 1-vs-3 and adapts.

can be seen by noting that although the performance is low in the first $\approx 5$ steps, after $\approx 30$ steps the model rapidly achieves the baseline *lower bound* performance, corresponding with continuous training without any distractor tasks. It is noteworthy that (1) increasing the number of distractor tasks from 5 to 15 has no effect on the speed at which the network achieves baseline performance (no negative transfer), and moreover (2) the introduction of distractor tasks actually *improves* the final model performance. This observation implies positive backward transfer; the model has learnt partial solutions from distractor tasks that can be composed to construct a better solution to the initial task.

**Positive Forward Transfer.** A key challenge in continual learning is to effectively leverage solutions to previous source tasks to achieve more data efficient learning of an unseen target task. NCTL is explicitly designed to support forward transfer by identifying locally relevant solutions to form a good initialization without any knowledge of task information, by performing Bayesian model averaging over temporal segmentations and previous model states. To assess forward transfer in a continual learning setting, we evaluate the efficiency with which NCTL is able to solve a randomly sampled target task having been trained on varying numbers of source tasks, averaging over 500 seeds. It is evident in Figure 3 that performance improves monotonically in the number of source tasks, even though conflicting tasks are permitted and more likely to exist with larger number of source tasks.

**Performance versus Task-Specific Oracles.** We now investigate the overall learning behaviour of NCTL on continual stream of Free Split MNIST tasks. Figure 4 shows the performance of NCTL compared to two oracle baselines averaged across 1000 Free Split MNIST task sequences. Oracle 1 has the benefit of being aware of task boundaries (but not task identities) and instantiates a new GLN upon each task change in order to prevent negative transfer. Oracle 2 has full knowledge of both task boundaries *and* task identities, and is therefore able to restore and continue training a GLN associated with previous instances of this task. Both oracles used GLNs with exactly the same architecture and hyper-parameters as NCTL. Unlike Oracles 1 and 2, NCTL is at a disadvantage as it is not informed of the task identities or task boundaries. Despite this, NCTL significantly outperforms Oracle 1. It further succeeds in outperforming Oracle 2, which has a dedicated model per task. This phenomenon can only be explained by NCTL exhibiting positive forward transfer across different tasks.

**Transfer Interpretability.** NCTL has the desirable property that task-specific solutions can be visualized directly. This property follows from the multi-linear structure of the underlying GLN model, i.e. the inference function collapses to a data-dependent multi-linear polynomial of degree equal to the number of GLN layers [VLB+19]. This same technique can be directly applied to the network of maximum a posteriori (MAP) GGM solutions under each local FMN process. This natural formulation of a saliency map can be further applied to visualize learning and knowledge transfer, as demonstrated in Figure 5.

As an additional validation that pseudo-tasks formed by NCTL from local neuron ensembling lead to meaningful transfer, we inspect saliency maps of a small network with a single layer of 5 neurons

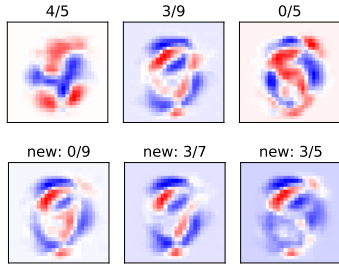

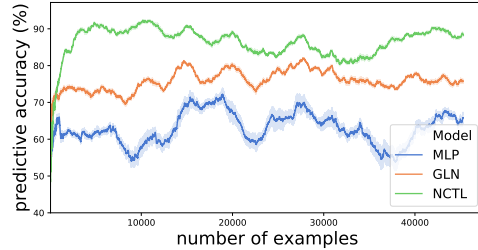

Figure 6: (**Top**) saliency maps for a tiny 6-node NCTL trained on 3 Free Split MNIST tasks: 4-vs-5, 3-vs-9 and 0-vs-5. (**Bottom**) saliency maps resulting from local ensembling of these nodes, which form useful initializations for the unseen tasks 0-vs-9, 3-vs-7 and 3-vs-5.

Figure 7: NCTL performance on real-world *Electricity* pricing online prediction problem, compared to GLN [VLB$^+$19] and MLP. NCTL and GLN are trained online, MLP is trained with batch size 20. Error bars denote 95% confidence levels over 10 random model initializations.

aggregated by a single output neuron. Figure 6 shows that, after training on 3 tasks (4-vs-5, 3-vs-9 and 0-vs-5), randomly sampled pseudo-tasks not only preserve solutions for source tasks but also include qualitatively useful instantiations for unseen target tasks.

## 5.2 Performance Benchmarking

In this Section we evaluate NCTL compared to previous state-of-the-art algorithms on three standard benchmarks: (1) Disjoint Split MNIST, discussed in the previous subsection; (2) Disjoint Split Fashion MNIST, which shares the same splitting protocol as Disjoint Split MNIST but with Fashion MNIST as the underlying dataset; and (3) Permuted MNIST [GMX$^+$13, KPR$^+$17], where tasks are defined by a choice of pixel permutation which is consistently applied to each image in a given task. NCTL was implemented using JAX [BFH$^+$18] and the DeepMind JAX ecosystem [BHK$^+$20, HCNB20, HBV$^+$20, BHQ$^+$20]. Code at: `github.com/deepmind/deepmind-research/`.

We consider a *domain-incremental* [vdVT18] continual learning scenario assuming known task boundaries. Task identities are available during training but not testing so that "multi-headed" solutions (each task maintains its own output units) are not applicable. We put NCTL at disadvantage compared to the previous methods by providing it with neither task boundaries or identities. We first evaluate on Disjoint Split Fashion MNIST where forward transfer and resilience to catastrophic forgetting can be compared throughout the training process. We then compare the final accuracies of NCTL to a broad set of continual learning methods for Permuted and Split MNIST variants.

**Split Fashion MNIST.** We compare performance of NCTL to two popular continual learning algorithms: Elastic Weight Consolidation (EWC) [KPR$^+$17] and Online EWC [Hus18, SLC$^+$18]. We also present the performance of GLN [VLB$^+$19] which shares the same network structure as NCTL and has been shown to be robust to catastrophic forgetting. The main difference between EWC and Online EWC is that Online EWC is more time and space efficient as it uses a running sum of task-specific Fisher information matrices instead of one matrix for each task. Both algorithms were trained in the same batch (not online) regime described by the original authors [KPR$^+$17], with additional access to information regarding task boundaries and identities. We used MLP with layers consisting 1000-250-1 neurons, ReLU nonlinearities between the first two layers and sigmoid applied to the final layer. Hyper-parameters are optimized by grid search for both EWC and online EWC: the regularization constant $\lambda$ is set to $10^6$ and the learning rate is set to $10^{-5}$ for EWC; and we have $\lambda = 10^7$, a learning rate of $10^{-5}$, and the Fisher information matrix leak term $\gamma$ (based on the formalism of [SLC$^+$18]) set to $0.8$ for online EWC. Our NCTL consisted of 50-25-1 neurons where the base model for each neuron is a GGM with context space $C = 2^4$ trained with learning rate 0.001. We adopted the same hyperparameters for the GLN baseline.

Figure 8 shows the test accuracy on Split Fashion MNIST while being trained in a single pass and evaluated every 500 examples. As described above, NCTL does not use one set of parameters but rather leverages local solutions to quickly adapt and instantiate different solutions for different tasks. When evaluating accuracy for NCTL on any task we therefore provide it with a small amount of data

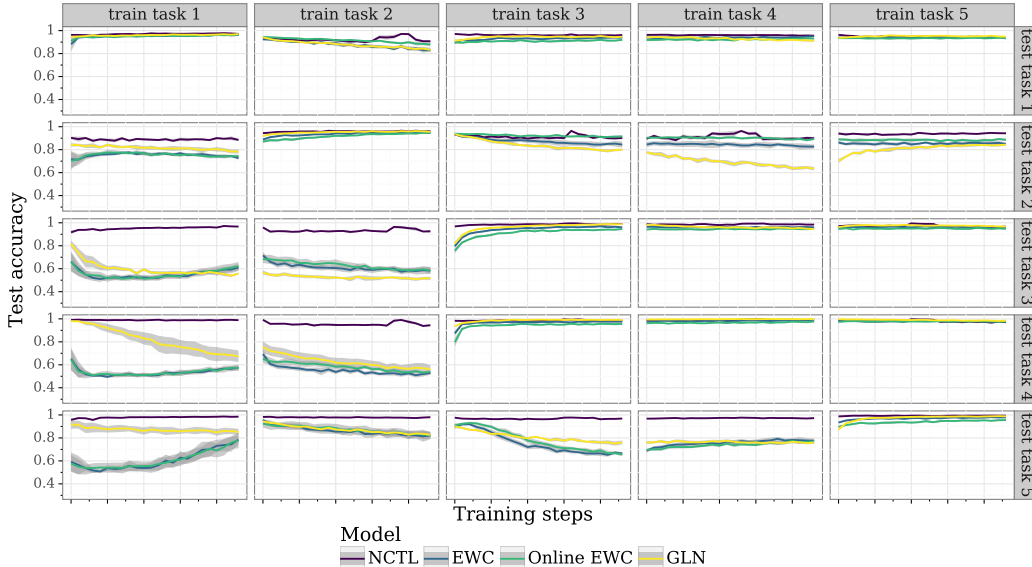

Figure 8: NCTL forward and backward transfer performance for Split Fashion MNIST, compared to Elastic Weight Consolidation (EWC) [KPR+17], Online EWC [Hus18, SLC+18] and GLN [VLB+19]. Models are trained sequentially on up to 5 tasks (rows) and evaluated on each of these tasks (columns), e.g. the top-right plot indicates performance on Task 1 after being trained sequentially on Tasks 1 to 5 inclusive. Each model only trains for one epoch per task. Error bars denote 95% confidence levels over 10 random seeds.

(50 examples) for adaptation before evaluation. We emphasize that this amount of data is insignificant to contemporary batch i.i.d methods. Therefore, we also included a GLN baseline, which is well suited to fast, online learning and is known to be resilient to catastrophic forgetting [VLB+19]. The rows of Figure 8 represent train tasks and the columns represent test tasks, e.g. the top-right panel compares performance on task 1 for models pre-trained on tasks 1 through 4 and being trained on task 5. It is evident that NCTL matches or exceeds performance of both EWC variants on all combinations. This is particularly evident in the lower half of the plot, i.e. NCTL is able to rapidly adapt to all tasks 2 through 5 having only been pre-trained on task 1, and the performance remains constant at worst, whereas other baseline performances often degrade. This demonstrates positive or at least non-negative forward transfer for NCTL which is a rare feat, especially without accessing task boundaries/identities. The upper half of the plot shows that the resilience of NCTL to catastrophic forgetting is comparable to EWC and Online EWC, if not sometimes better.

**Split and Permuted MNIST.** Here we use the same NCTL configuration of Split Fashion MNIST for Split MNIST. For permuted MNIST we construct a one-vs-all classifier with the same un-optimized setting as Split Fashion MNIST, but with smaller networks consisting 10-5-1 neurons only and context space $C = 2^6$. Table 1 compares NCTL to the suite of continual learning methods benchmarked in [HLRK18]. It is clear that NCTL significantly outperforms all previous methods that do not make use of an explicit replay mechanism: GEM [LPR17] stores fixed amount of examples per prior task to provide constraint when learning new examples; DGR [SLKK17] and RtF [vdVT18] are state-of-the-art rehearsal-based methods with generative model. The performance advantage is particularly significant for Split MNIST, where the gap to the next best method is 24%.

### 5.3 On Sources with Unknown Drift

The *Electricity* (Elec2-3) dataset [HW99] contains 45,312 instances collected from the Australian NSW Electricity Market between May 1997 and December 1999. Each instance consists of five attributes and one binary class label indicating direction of price movement relative to the past 24 hours. There are two attributes for time: day of week and period of day. The remaining three numeric attributes measure current demand: the demand in New South Wales, the demand in Victoria, and the scheduled transfer between the two states. As the data is derived from real-world phenomenon, instances are subject to continuous drift of underlying data distribution, and we cannot know definitely

Table 1: Average domain incremental Split/Permuted MNIST accuracies of NCTL versus a benchmark suite of continual learning methods, using the setup described in [HLRK18]. NCTL significantly outperforms all other replay-free algorithms, and is able to achieve comparable performance with the replay-augmented GEM, DGR and RtF algorithms. Note that NCTL alone is solving a strictly more difficult variant of the problem where task boundaries and identities are not provided.

| Method | Replay | Task Boundaries | Split MNIST | Permuted MNIST | Reference |
|---|---|---|---|---|---|
| **NCTL** | | | **95.07** $\pm$ 0.02 | **95.27** $\pm$ 0.00 | [ours] |
| EWC | | ✓ | 58.85 $\pm$ 2.59 | 91.04 $\pm$ 0.48 | [KPR$^+$17] |
| Online EWC | | ✓ | 57.33 $\pm$ 1.44 | 92.51 $\pm$ 0.39 | [Hus18] |
| SI | | ✓ | 64.76 $\pm$ 3.09 | 93.94 $\pm$ 0.45 | [ZPG17] |
| MAS | | ✓ | 68.57$\pm$ 6.85 | 94.08 $\pm$ 0.43 | [ABE$^+$18] |
| LwF | | ✓ | **71.02** $\pm$ 1.26 | **72.64** $\pm$ 0.52 | [LH17] |
| GEM | ✓ | ✓ | 96.16 $\pm$ 0.35 | 96.19 $\pm$ 0.11 | [LPR17] |
| DGR | ✓ | ✓ | 95.74 $\pm$ 0.23 | 95.09 $\pm$ 0.04 | [SLKK17] |
| RtF | ✓ | ✓ | **97.31** $\pm$ 0.11 | **97.06** $\pm$ 0.02 | [vdVT18] |
| Offline (upper bound) | | | 98.59 $\pm$ 0.15 | 97.90 $\pm$ 0.09 | [HLRK18] |

if or when such drift occurs. This presents challenges to existing methods relying on knowledge of task boundaries and/or task identities.

We evaluate NCTL compared to GLN [VLB$^+$19] and MLP on this task by processing examples in temporal order, with each model first making a probabilistic prediction and then updating its weights. Here we use the same configurations as detailed in Split Fashion MNIST, with learning rate of 0.1 for NCTL and GLN, and $10^{-4}$ for the MLP. Figure 7 shows superior performance of NCTL with an overall accuracy of 86.37%. For reference, an online version of C4.5 [Qui14] called SPLICE-2 [HW99] reports accuracy between 66% and 67.7% using decision tree built from examples in sliding window of varying sizes. Later work [KM07] reports accuracy of 62.32% for naive Bayes and 80.75% for DWM-NB which is an ensemble method that dynamically creates and removes experts in response to changes in performance, using an incremental version of naive Bayes for the ensemble components.

# 6 Future work

In this work we have focused on small scale binary classification tasks that are well-established in the continual learning community. The NCTL algorithm can be naturally generalized to other online settings such as multiclass classification and regression by using more general forms of geometric mixing [Mat13, BMS$^+$20], and it would be interesting to apply the algorithm to these settings. Perhaps the most promising future application area for this work is reinforcement learning, however it is as yet unclear how to best combine the local learning mechanism in NCTL with bootstrapped targets in a principled fashion.

# 7 Conclusion

In this paper we described a framework for combinatorial transfer, whereby a network with $m$ nodes trained on $h$ tasks generalizes to $h^m$ possible task instantiations (pseudo-tasks). This framework relies on the ability to meaningfully ensemble networks at the level of individual nodes, which is not a property of contemporary neural networks trained via back-propagation. Instead, we provide a concrete instantiation of our framework using the recent and complementary Forget-Me-Not Process and Gated Linear Network models. We provide a variety of experimental evidence that our NCTL algorithm does indeed exhibit combinatorial transfer, and that leads to both positive forward and backward transfer (no catastrophic forgetting) in a difficult online setting with no access to task boundaries or identities. Moreover, we demonstrate empirically that the distribution of pseudo-tasks is semantically meaningful by comparing NCTL to a number of contemporary continual learning algorithms on standard Split/Permuted MNIST benchmarks and real-world *Electricity* dataset. This new perspective on continual learning opens up exciting new opportunities for data-efficient learning from a single temporal stream of experience in the absence of clearly defined tasks.

## Broader Impact

This paper introduces a novel combinatorial perspective on transfer learning: past experiences can generalize to an exponentially growing number of unseen tasks. Taken to its limits, an obvious widespread benefit is improved data efficiency in solving unseen tasks, which could dramatically reduce compute and energy consumption.

Privacy and algorithmic bias should be considered for real world applications to ensure that they are ethical and of positive benefit to society. Our proposed algorithm is online and therefore does not necessarily require storing data, which is potentially beneficial in terms of privacy. The algorithm inherits many interpretability properties from GLNs [VLB$^+$19], which might be helpful for understanding and addressing any potential bias issues in deployment.

## Funding Disclosure

All authors are employees of DeepMind.

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
