[Reviews · NeurIPS 2020]

Review 1

Summary and Contributions: The authors extend the GLN model with the Forget-Me-Not-Process to allow effective bayesian model averaging and for use in “online continual learning”. Results are provided for several (toy) binary benchmark tasks.

Strengths: - The authors propose an interesting idea to achieve transfer by ensembling - The authors build on an interesting and novel direction of leveraging backprop-free algorithms to achieve more efficient online and non-stationary learning - The performance on some of the experiments is promising

Weaknesses: -Experiments are Toy, which is not uncommon for this setting. However, for this particular method since it uses a newly introduce backprop free alternative reliant on a special architecture I am particularly concerned at the scalability and power of the GLN on realistic datasets. The prior work and this one does not establish that GLN can perform well on problems where Deep Networks are critical, e.g. CIFAR-10, imagenet, etc. MNIST and Permuted MNIST can yield strong results even with a single hidden layer. The use of a different architecture in the comparisons particularly concerns me, I understand it is needed to apply GLN, but what if the competing methods are applied to a different architecture e.g wide 1-hidden layer network for example on these toy problems. -Why is the method not compare to the original GLN paper? Some of the experiments are very similar looking. -EWC is used as the key comparison but EWC is well known to perform poorly in the one-pass through the data/online CL setting [a][b] -The authors delineate DGR as using “Experience replay”, however this model doesnt actually store prior data (besides prior models similar to NCTL). I think a more appropriate delineation is related to the size of the storage. -Rehearsal based ER has been shown to be as effective or outperforming GEM [a][b] -Can the authors comment on how to extend the model to non-binary case -Theoretical grounding - The idea to achieve combinatorial transfer visa ensembling seems great but it is not 100% clear theoretically why the method should achieve combinatorial transfer via ensembling. Some more, intuition at least, regarding this would help the paper. [a] Chaudhury et al . On Tiny Episodic Memories in Continual Learning Chaudhury. [b] Aljundi et al. Online Continual Learning with Maximally Interfered Retrieval.

Correctness: Yes

Clarity: I find the writing unclear. Particularly in the description of the main ideas of the algorithm and the distinction and delineation to the exact contribution of this work with respect to the prior work on GLN, this could be emphasized. The ideas about the importance of modularity in continual learning are well known in the continual literature, achieving them via model averaging is an interesting idea. The figure and intuition of what is desired (per node ensembling) is nice but it’s not completely clear to the reviewer the high level intuition of why effective model averaging at nodes as shown in the Figure is achieved in this work. I would ber really interested in more intuition or arguments from the authors about why the ensembling works for GLN. Are the authors exploiting local convexity of the subproblems which allows to claim a global optima? Finally it might be a good idea to include a full algorithm block particularly one that illustrates the FMN process and when it is applied in the supplementary materials at least.

Relation to Prior Work: This could be improved as mentioned above. In addition I note there is many recent works on "online continual learning" e.g. [a,b] but also many others. The authors should at least review this more thoroughly. [a] Chaudhury et al . On Tiny Episodic Memories in Continual Learning Chaudhury. [b] Aljundi et al. Online Continual Learning with Maximally Interfered Retrieval.

Reproducibility: Yes

Additional Feedback: Reproducibility There is no publicly available code for GLN although some basic hyperparameters are given in the paper. In addition the FMN is somewhat complex. It is the reviewers belief that it would be very challening to reproduce the results here. Post rebuttal: I have read the rebuttal, the authors have addressed some of my concerns. However as mentioned in my review it is still not clear why the node level averaging should be meaningful. For example would there be any difference in results if one permuted the nodes of one of the models at a given layer before ensembling? Since the experiments are relatively toy, I think its important to more thoroughly motivate the methods potential. Overall The direction of GLN and backprop free methods for addressing online and continual learning is very interesting. The work is promising but currently the experiments are not completely convicing to the reviewer (especially the lack of comparison to naive GLN in Fig 7) and the writing really seems like it could use another iteration.


Review 2

Summary and Contributions: The paper proposes a novel method for performing positive forward and backward transfer for continual learning. The key idea is to combine the Forget-Me-Not process and Gated Linear Networks to perform task segmentation, modular learning and memory-based ensembling to enable generalization on a growing number of unseen tasks sequentially.

Strengths: The major strength of this work is that the algorithm has been designed from the perspective of handling task sequences containing non-stationary data distributions at its core, while at the same time being agnostic to task boundaries and task identities.

Weaknesses: Please see my comment below.

Correctness: Yes

Clarity: The descriptions of GGM, FMN and NCTL are quite terse to understand and need to be re-read a couple times to make sense of them. I'd recommend simplifying these descriptions for an easier flow and deferring the details to an appendix. Other than this, the rest of the paper is mostly well-written and easy to go over.

Relation to Prior Work: Yes

Reproducibility: Yes

Additional Feedback: My major concern is that the authors have only applied their method to variants of MNIST. While the experiments performed are indeed from the established Continual Learning benchmarks in prior work, they do not suffice to showcase the true complexity of the continual learning challenge. I would strongly recommend doing at least some RL experiments, for instance, as performed in Online EWC paper. Secondly, as mentioned above the descriptions of GGM, FMN and NCTL are quite terse to understand and need to be re-read a couple times to make sense of them. I'd recommend simplifying these descriptions for an easier flow and deferring the details to an appendix. --- Post-rebuttal comments --- After going over the author response, my score hasn't changed so far. The authors have not addressed any of my comments in the response about applying their method to continual learning in the RL setting. Other reviewers have also commented on the scalability of their setting and the authors have mentioned that there might exist potential extensions of their method to ImageNet and CIFAR datasets, by incorporating convolutional inductive biases (although it might be complicated to do so and they do not mention how this could be done). While this could involve some work to figure out how these inductive biases can be incorporated, I believe RL experiments on small scale should be possible. They need not use visual input, rather one could work in a quadruped setting and use joint angles, joint velocities and potentially some food sensing (e.g., using a vector directed towards the closest food pellet) as inputs which would then enable them to use their approach to learn a sequence of tasks, e.g., (a) learning to get up, (b) learning to stand, (c) learning to crawl, (d) learning to walk, (e) learning to jump, (f) gathering food, (g) learning to run, (h) collecting maximum food pellets in a limited time and so on. So currently, I think I'll stick to my score of 6 due to a lack of thorough experimentation in diverse settings.


Review 3

Summary and Contributions: The authors introduce a new way of tackling continual learning problems, by combining different modules learnt on previous data. They target the online continual learning setting (no task boundaries / identities provided to the system). They use Gated Linear Networks and the Forget-Me-Not process to realise an algorithm that performs well on MNIST datasets. They show some properties of the algorithm.

Strengths: This is a novel way to approach the continual learning problem. It targets online continual learning as opposed to other works that require task boundary and identity information. Therefore its significance is also high. It is relevant to NeurIPS. The paper is well-written and structured. I would ask for more details on GGMs and the FMN process in an Appendix with the authors' notation (instead of pointing towards the relevant past work). The overall algorithm is motivated well intuitively (with experiments showing this in Section 5.1) and the theoretical grounding seems justified. Figure 1 is nice. The experiments on Split MNIST and Permuted MNIST show good results, with the competing algorithms requiring more information than NCTL, and even then not outperforming. ---------------- After author response: Thank you for the author response. I will keep my score, and recommendation of accept. - The clarification on Oracles was useful. I also think some discussion on the complexity of the algorithm is very important in a future version of the paper. - It is my opinion that the experiments in this paper are good enough for a NeurIPS paper. Of course, as other reviewers say, if the authors were to provide more diverse (eg RL) or larger scale experiments, that would make this paper extremely strong. - R1's comments about further intuition as to why node-level averaging works is very interesting. I think this work's impact would be improved by exploring this question further. - Small point: Figure 7 is very hard to read. The quality of this Figure could be improved.

Weaknesses: The major weakness appears to be the scalability of this algorithm. Firstly, as the authors only use MNIST datasets, they can get away with small numbers of nodes (and do not require any 'convolutional' style elements). Secondly, the time and memory requirements grow with sample number (something like O(n log n) and O(log n)?). Both these mean I do not expect this algorithm to scale well to larger datasets. This is the main problem for me, despite me liking this idea. A simple discussion on the scalability should be enough for me (currently lacking in the paper). It might help to provide the raw training times on Split and Permuted MNIST for the algorithms. I am not necessarily asking for larger experiments (I believe the novelty of this idea and the experiments provided are enough). I do not understand why NCTL outperforms the Oracles in Figure 4. Don't the Oracles also have potential for forward transfer? Some more explanation (aside from the one sentence in line 216) would be good.

Correctness: The claims and method, to the best of my knowledge, are correct. The empirical methodology is correct and explained.

Clarity: The paper is well written and easy to read. Further details on parts of Section 4, in for example an Appendix, would be good.

Relation to Prior Work: Prior work is discussed and built upon in this work.

Reproducibility: Yes

Additional Feedback:


Review 4

Summary and Contributions: This paper proposes to use Gated Geometric Mixer and Forget-Me-Not Process to tackle the problem of continual transfer learning. GGM is used for node-level modular learning and FMN in intended for local ensembling. Extensive experiments are conducted on MNIST to show the validity of the proposed methods and comparison with previous continual learning algorithms.

Strengths: The proposed framework builds upon existing GGM and FMN and manage to ensemble networks at the level of individual substructure of neural networks, which is beneficial for transfer learning. The experiments conducted on offers a good insight how this algorithm helps in terms of combinatorial transfer.

Weaknesses: The biggest question I have is the generalization ability of proposed method. Current GGM and FMN deals with single binary event and it remains unclear how will they be adapted to multi-class classification task and how the computing time and space will change in FMN.

Correctness: The proposed NCTL can deal with catastrophic forgetting problem in transfer learning by doing experiments over positive backward transfer and positive forward transfer.

Clarity: The paper is well motivated and easy to follow. Even though I am not an expert on the topic, I could follow all steps including the help of provided references.

Relation to Prior Work: Yes.

Reproducibility: Yes

Additional Feedback: I'm wondering how the proposed methods will perform on complex tasks such as object detection or fact learning as in [ABE+18] or any insights over the generalization ability. Post rebuttal: I have read the rebuttal and some of my concerns are addressed by the authors. I'm changing my ratings to 6 with major concern remaining in the scalability of experiment settings. The ability to overcome catastrophic forgetting problem in continual learning is more desired in task execution settings, where the task boundaries are more separated to show the effectiveness of NCTL in positive forward and backward transfer. In this sense it can be better evaluated in RL settings as also posted by other reviewers. Overall the paper points out a interesting direction and more promising results are expected.

[Author Response · NeurIPS 2020]

We thank all of the reviewers for their time, effort and engagement with our work. Feedback is addressed below, and all comments will be incorporated in the final manuscript revision.

**=== General Comments ===**

**Extension to non-binary targets.** To deal with different settings, note that gated geometric mixing neurons can be defined for any member of the exponential family, not just the Bernoulli case as considered in this work. This includes both the Categorical distribution and the Gaussian distribution, which enables them to perform multi-class classification and regression respectively. A concrete example of this is the "Gaussian Gated Linear Networks" paper (which can be found on arXiv) that shows SOTA results on many regression problems. In the categorical case, the geometric mixture probability of a class $c$ in $\{1, 2, \ldots, C\}$ given $m$ categorical distributions $p_i(.)$ with weight $w_i$ for $i = 1..m$, is $\Pr(c) := \exp\{\sum_{i=1}^{m} w_i \log p_i(c)\} / \sum_{c'=1}^{C} \exp\{\sum_{i=1}^{m} w_i \log p_i(c')\}$, which requires an additional factor of $C$ work (like softmax) per neuron, and no additional space complexity. NCTL is orthogonal to this specific choice of neuron parameterization, but we elected to focus on classification tasks that are well-established in the continual learning community.

**On scale of datasets.** We agree that the continual learning problem is far more complex than captured by current standard datasets. We elected for the datasets that allowed us to make the most direct comparison to existing SOTA methods. We agree that these datasets are small compared to those considered in large-scale representation learning (e.g. ImageNet), but it's worth noting that (1) the two fields are solving very different problems, and (2) that even MNIST variants are sufficiently complex to clearly stratify the performance of competing methods (the function of a challenge dataset). This largely indicates that online/continual learning is an under-developed field compared to large-scale representation learning (ImageNet) or language modelling, and currently there is little advantage (but much cost) to running experiments at this scale. That said, we can imagine several ways in which GLNs could be augmented e.g. with convolutional inductive biases to scale to larger problems, and agree this is necessary for ImageNet-style problems, but this would be a paper in its own right and is orthogonal to what we were aiming to achieve with this work.

**On clarity of exposition.** Thank you for all the helpful suggestions, they are well taken. If accepted we will endeavor to use the additional page of space to properly address the additional related work mentioned, add further details to the appendix and expand the discussion in the *broader impact* statement.

**Specific Comments (not addressed above):**

**=== Reviewer 1 ===**

**Comparison to EWC.** This is a misunderstanding so we thank the reviewer for raising it. EWC was trained in the same batch (not online) regime described by the original authors, with additional access to information regarding task IDs and boundaries. NCTL has access to none of this information, and is the only method that has been trained subject to these additional constraints. The simple reason is that NCTL is the first method (to our knowledge) that can operate in this strictly more difficult regime. We will clarify and emphasize this critical point in text.

**Why is the method not compared to the original GLN?** The only similar experiment in the original GLN paper is the evaluating of robustness to catastrophic forgetting (negative backward transfer), owing to the simhash-like inductive bias induced by half-space gating. We can add this baseline to the next revision but note that vanilla GLNs have no forward-transfer properties like NCTL, which is the more difficult direction addressed in this work.

**Source code and reproducibity.** One can find independent reimplementations of GLNs and FMN online. We will open source our implementation in time for NeurIPS. Unfortunately we were unable to obtain the necessary approvals in time for submission/review, but this has since been corrected.

**=== Reviewer 3 ===**

**On Scalability.** There is a slight misunderstanding regarding the asymptotic time complexity of the algorithm. Assuming a fixed network structure and a constant sized memory pool of $k$ items, the per example running time of NCTL is $O(k \log n)$. To process $n$ symbols, the time complexity is $O(nk \log n)$ with a space overhead of $O(k \log n)$. We will clarify this further in the next revision of the paper.

**Why does NCTL outperform the Oracles in Figure 4. Don't the Oracles also have potential for forward transfer?** Consider this training setup: 1000 steps on Task 1, 1000 steps on Task 2, and another 1000 steps on Task 1. The stronger Oracle maintains two networks, one trained on 2000 steps of T1 and the other on 1000 of T2. NCTL (using the exact same underlying network/params) is exposed to all 3000 steps in an unlabelled stream, thus the only way it can outcompete the Oracle on T1 is to leverage information from the 1000 steps on T2. By definition this demonstrates positive transfer. We will clarify our explanation.

[Meta-Review · NeurIPS 2020]

This paper studies continual learning that does not require task boundary and identity information and proposes a novel model ensemble method from the combinatorial perspective for this problem. All reviewers and AC agree that this paper builds a novel and promising direction. Authors also design delicate algorithm by introducing the non-stationary learning techniques to solve this problem. The experimental results of this method are somewhat weak in several aspects, but given the challenge of online continual learning in nature, they are fairly convincing to justify the main ideas and proposed methods. Note that after rebuttal and discussion phases, there still remain several major concerns: First, the empirical evaluation is not realistic in terms of task diversity and scalability. Second, the explanation of the method still lacks clarity and the paper is not easy to follow. In particular, there is not a clear reason the node level averaging should be meaningful. Since the rebuttal only partly addresses the above comments, the authors are required to improve their paper dedicatedly following these comments. I recommend to accept this paper, assuming that the above major concerns will be addressed fully in the camera-ready version.